# Polytrauma Caused by a Bear Attacking a Human with a Benign Outcome

**DOI:** 10.3390/healthcare12050542

**Published:** 2024-02-24

**Authors:** Ruslan Mellin, Ellina Velichko, Larisa Maltseva, Sergey Dydykin, Yuriy Vasil’ev

**Affiliations:** 1G. Ya. Remishevskaya Republican Clinical Hospital, Abakan 655012, Russia; ruslanmellin@mail.ru; 2Pathological Physiology Department, I.M. Sechenov First Moscow State Medical University, Moscow 119435, Russia or linavel83@gmail.com (E.V.);; 3Operative Surgery and Topographic Anatomy Department, I.M. Sechenov First Moscow State Medical University, Moscow 119435, Russia

**Keywords:** bear attack, polytrauma, maxillofacial deformities, maxillofacial trauma, maxillofacial reconstruction surgery

## Abstract

Injuries to humans caused by wild animals, particularly bears, are rarely mentioned in the literature. Such injuries are frequent in Siberia, which is a territory surrounded by dense forests inhabited by brown bears. In the last 4 months alone (September–December 2023), four bear attacks on humans were registered in Khakassia, Russia. This article presents a clinical case of rehabilitating a patient after a bear attack, who suffered multiple fragmentary fractures of the facial skeleton with displaced bone fragments, subcutaneous emphysema of the soft tissues of the face, damage to the parietal and right occipital regions and paranasal sinus hemorrhage on the left side. The nature of the injuries was enhanced by trauma to the upper extremity caused by the patient defending himself against the animal. In addition to the damage to his face, the bear tried to open his cranium, as evidenced by four furrows caused by its canines, including two each on the frontal and occipital bones of the skull. The patient’s complex treatment included both maxillofacial and reconstructive surgeries, and outpatient treatment involved the formation of normotrophic scars using a neodymium laser and injections of a heterogeneous composition consisting of microparticles of “crosslinked” collagen of animal origin placed in a gel identical to the natural extracellular matrix.

## 1. Introduction

The anthropogenic impact on the habitats of especially large carnivores leads to less-stable conditions for their prey base. This has a negative impact on people in the vicinity of the habitats of large predatory mammals. Attacks by large wild animals on humans do not only occur during chance encounters in the wilderness. According to the observations of various authors, the most dangerous animals for humans belong to the families Canidae, Felidae, and Ursidae [1]. According to https://worldanimalfoundation.org/ (accessed on 18 February 2024), globally, the average number of annual bear attacks on humans is 39.6, of which 11.4 and 18.2 attacks occur in North America and Europe annually. In Scandinavian countries, according to Støen O.G. et al., the brown bear (Ursus arctos) population increased from ~500 individuals in 1977 to ~3300 in 2008, leading to an increase in injuries, fatalities and public fear of bear attacks [2,3]. In the 2000s, 46 bear attacks, resulting in 48 fatalities, were reported in North America. Of these, there were 25 and 21 respective fatal attacks caused by black and brown bears. Of these incidents, 19 occurred in Canada, and 27 occurred in the United States. In Alaska, there were eight fatal bear attacks involving 10 people between 2000 and 2017. During the same period, the proportion of accidents in Alaska out of all the fatal attacks in the U.S. was 29.6%, which represents 17.4% of all the fatal attacks in North America. The proportion of bear fatalities in the state of Alaska out of the U.S. is 34.5%, and out of all the bear fatalities in North America, this is 20.8% [4]. In Russia, according to statistical data from the Komi Republic, 98 human encounters with brown bears were registered in 2019 alone. During 1999–2020, in this republic, bears attacking humans (wounded/killed) represented 4.8% of the bear activities in populated areas. At the same time, no unified statistics on accidents caused by human–bear encounters were found in the literature [5,6]. In China, 1.28% of the cases in which bears inflicted harm resulted in human injuries [7]. In Russia, Canada, and the United States, the bear–human conflict remains highly tense [1,2,3,4,5,6], manifested not only by human deaths but also by severe combined injuries complicated by shock, septic inflammation, and the risk of rabies and tetanus infection [8,9,10,11]. As recommended by the WHO, patients attacked by a bear are vaccinated against rabies and injected with tetanus anatoxin [12].

There are extracts from patient records in the literature that can be used as guidelines for treatment and rehabilitation in similar cases. According to Floyd T., the predominant oral fluid flora of black bears is sensitive to penicillin [13]. Vishal et al. (2022) report that it is important to perform initial surgical debridement, and work on deep defects, including injuries of both soft tissues and bones, should be carried out with both the patient’s own tissues from the wound zone and artificial biocompatible scaffolds [12].

The goal of this article is to present the case of a bear attacking a person in which the victim survived, as well as to analyze the literature data on the prevalence of attacks and the procedure for providing medical care.

## 2. Materials and Methods

Patient S., who was 42 years old, was taken by air ambulance to the G. Ya. Remishevskaya Republican Clinical Hospital’s (Abakan, Russia) maxillofacial and plastic surgery department.

### 2.1. Case History

According to the reports of the air ambulance doctor and the victim, the latter was injured in the forest. One morning in September 2023, a wild bear attacked a person while he was harvesting pine nuts. According to the patient, first, the bear gnawed through his left hand, which he used to cover his face; then, it pressed his body to the ground with a paw and started gnawing his face. Then, the animal tried to open the cerebral part of the cranium, as evidenced by four furrows caused by the canines, two each on the frontal and occipital bones of the skull. The man remained conscious during the fight. After the victim stopped resisting, the bear left him. After waiting for about 10 min, he rose on his own, made sure that there was no predator nearby, and went to the side of the camp. A helicopter rescue service was called. After 9 h and 30 min, the patient was taken to the reception department of the Republican Clinical Hospital of Abakan. Spiral computed tomography of his whole body was performed. The patient was examined by an on-duty intensive care physician (traumatic hemorrhagic shock was diagnosed), traumatologist (open fractures of the left upper limb were diagnosed), neurosurgeon (craniocerebral trauma was present), general surgeon (internal organ trauma did not occur, but there were multiple bite wounds on the trunk and extremities) and maxillofacial surgeon. Due to the bones and soft tissues of the maxillofacial region being significantly damaged, the patient was hospitalized in the department of maxillofacial and plastic surgery. The patient was prophylactically injected with the following vaccines: concentrated, purified, inactivated antirabic culture vaccine (COCAV) (Microgen, Russia) and purified adsorbed liquid tetanus anatoxin (AC-anatoxin) (Microgen, Russia). The patient signed an informed written consent, which included permission to publish their medical history data and photo protocol.

### 2.2. Objective Examination Findings

External examination revealed facial configuration disorder due to multiple bite wounds on the frontal and periorbital regions on both sides; the suborbital, zygomatic, and nasal wings on the right side; the parotid region on the left side and upper and lower lips; and the occipital region. The wounds were contaminated by sand, hay, pine needles, grass, leaves, and dead insects (Figure 1). In the buccal–zygomatic region on the right side, the wound had a flap and penetrated the oral cavity. At the bottom of the wound, the following elements were visualized: a partially disconnected facial nerve; the parotid salivary gland (the salivary duct of the gland was not visualized); mimic muscles with integrity disorders; fragments of the maxilla with teeth; and fragments of the zygomatic bone. The right zygomatic bone was broken; this splintered fracture presented with significantly displaced fragments. The wounded nasal region was also visualized, the bones of which were also broken. The wound in the left parotid region had a semi-lunar shape; the parotid salivary gland was injured at the bottom of the wound. In the oral cavity, the mucous membrane of the palate was torn into several parts. Due to contamination of the wounds, it was not possible to determine their direction. Both maxillary sinuses were connected to the oral cavity due to the detachment of the alveolar process. The alveolar and partially palatine processes of the maxilla lay loose on the tongue and were not connected to the mucous membrane of the maxilla. The frontal part of the alveolar process, together with teeth 1.1 and 2.1, was rotated and displaced to the soft tissue of the left cheek. The mucous membrane of the cheeks and the oral vestibule had multiple flap wounds that were exposed to the external environment (Figure 1A).

According to the spiral CT scan on 18 September 2023 (#12905), no foci of pathological density were detected in the brain substance. The lateral ventricles were not dilated or symmetrical. The sub-arachnoid space was moderately dilated. The medial brain structures were not displaced. The Turkish saddle area was unchanged. The right zygomatic arch was fractured and displacement at an angle. The alveolar process of the maxilla on the left side incurred a multifocal fracture, with displaced bone fragments on the inside, and a downward and multifocal fracture of the anterior, inferior, and lateral walls of the right maxillary sinus, with bone fragments displaced both laterally and up to 18 mm into the sinus cavity. The lower wall of the left maxillary sinus (hemorrhagic substrate up to 5 mm thick in the cavity) was fractured. The nasal bones incurred a splinter fracture and were displaced. The coronary process of the mandible on the left side was fractured and moderately displaced. A large amount of free gas was detected subcutaneously in the soft tissues of the face on both sides, with significantly more on the right side, right occipital, and both parietal regions (Figure 1B). No bone traumatic pathology was detected in the cervical and thoracic spine. Degenerative/dystrophic changes in the cervical and thoracic spine were found. Using chest X-ray, it was found that the lungs were spread, and there were no focal and infiltrative shadows. The pleural cavities were clear. No traumatic bone pathology was observed via scanning. No free gas or fluid in the abdominal and pelvic cavities was observed using abdominal radiography. There was no traumatic pathology to the parenchymatous organs. Conclusion: There were multiple fragmentary fractures of the facial skeleton, with the displacement of bone fragments. Subcutaneous emphysema of soft tissues of the parietal and right occipital regions of the face was observed. There was a paranasal sinus hemorrhage on the left side.

The following diagnoses were made: 

There were multiple bite injuries to the face. The zygomatic bone on the right side was fractured and displaced. Traumatic amputation of the right and left sides of the maxilla occurred, causing type II and I Le Fort fractures, respectively. The nasal bones incurred a splinter fracture and were displaced. The face incurred multiple bite wounds without soft tissue defects. The left superior rectus muscle was detached. The Stenon’s duct on the right was ruptured. Paresis of the facial nerve on the right was indicated. The occipital region of the scalp was wounded. The trunk and extremities suffered multiple bite wounds. The lower third of the left humerus incurred an open multifocal fracture. The bases of 1–2 metacarpal bones on the left side incurred open fractures.

The lower third of the left humerus incurred an open comminuted fracture with displaced fragments. The diaphysis of the left ulna suffered an open fracture and was displaced. The first metacarpal bone on the left suffered an open comminuted fracture and was displaced.

## 3. Results

The first step was initial surgical debridement of the upper limb; Kirschner wires were used for temporary fixation to restore the axis of the left upper limb, and then plaster was applied for immobilization. The following foreign bodies were removed from the maxilla-facial region: leaves, pine needles, grass, teeth of the upper and lower jaws, and small fragments of maxilla bones (Figure 2).

The following intravenous antibacterial therapy was prescribed for 7 days: 2.0 mg Ceftriaxone and 30% Lincomycin solution. Due to good visualization of the right zygomatic bone through the existing wound, repositioning, and osteosynthesis with micro-plates and micro-screws manufactured by “Konmet” (Moscow, Russia) were performed. The displaced and rotated bones of the nasal dorsum were also placed in the anatomical position and fixed with the micro-plates and micro-screws (Figure 3).

Loose-lying fragments of the maxillary alveolar process were removed from the soft tissues of the face. Three of them were useful due to their large size and since they are anatomically important landmarks the subsequent reconstructive surgeries. These included two maxillary tubers and the frontal part of the alveolar process with teeth 1.1 and 2.1, which remained in it. The maxillary tubers were fixed on the micro-plates and micro-screws, with the anterior segment attached to two mini-screws (Figure 4). Excision of the crushed, in some places, necrotized, oral mucosa was performed. The paranasal sinuses were tamponaded through fenestration of the maxillary sinuses with the oral cavity using a nasal pack moistened with iodoform. The ends of the nasal packs were directed out through the artificially formed junctions in the lower nasal passages from both sides. Plastic surgery of the buccal mucosa was performed so that the flaps completely covered the fixed fragments of the maxilla. In the frontal section, the fixed bone fragment remained open when the surviving mucosas were brought together. The diastasis of the mucous membranes was about 1.0 × 1.7 cm in length. A nasal pack moistened with iodoform was sutured onto the open bone area. The wounds of the lips were sutured, which made it possible to separate the oral cavity from the external part of the wound.

In the initial surgical debridement of the facial wounds, a 0.7 cm long wound was found around the left upper eyelid. Within this wound, there was a formation dangling outward that looked like a tendon with a muscle. Having examined the muscle, pulling its outer end caused the pupil of the eye to turn downward (Figure 5). The stump of the left eye’s superior rectus muscle was fixed to the tarsal plate of the left upper eyelid. The skin was sutured with interrupted Surgic Pro 4/0 stitches.

When examining the patient on the first day after surgery, their eyes moved in all directions, symmetrically.

The following processes took 2/3 of the total operation time. The capsules of the parotid glands, muscles, and subcutaneous fatty tissue were sutured layer by layer with Vicryl 4/0. Interrupted Surgic Pro 4/0 stitches were applied to the skin (Figure 6). The wounds were drained with glove tapes. During the operation, 686 mL of erythrocyte mass and 580 mL of fresh frozen plasma were transfused into the patient.

To monitor the patient’s vital functions, the correct volume of erythrocyte mass (1350 mL) was transfused during their stay in the Department of Anesthesiology and Reanimation (DAR) to treat posthemorrhagic anemia and control the urinary function of the kidneys after surgery; then, the patient was transferred to the DAR on an artificial pulmonary ventilation machine. On the morning of the next day, he was taken off the ventilator to breathe independently through a naso–tracheal tube, and later that day, extubation was performed according to the indications. A naso–gastric tube was installed in the patient for nutrition. To prevent the development of salivary fistulas, atropine sulfate solution (0.1%, five drops given 30 min before oral intake three times a day) was administered to inhibit the functions of the salivary glands.

On the eighth day after the patient was stabilized and normal renal excretory function was restored, he was transferred from the DAR to a specialized department. To prevent the infection of the oral cavity wounds, the patient was allowed to eat through a naso–gastric tube. During postoperative treatment, the patient developed salivary fistulas in the parotid salivary gland wounds, which closed independently on the 15th day after the injury. Atropine sulfate was no longer administered. An intracanal fistula was treated using Bitodin solution, with a positive result. Due to the divergence of sutures in the projection of the alveolar process of the maxilla on both sides and to prevent the formation of oro-antral fistulas on the 16th day, repeated plastic surgery of the oral mucosa was performed under endotracheal anesthesia; the surgical sutures were also closed with nasal packs moistened with iodoform. The nasal packs were sutured to the oral mucosa. The previously installed nasal packs were removed from the maxillary sinuses. On the 23rd day, the patient was discharged in a satisfactory condition for outpatient treatment (Figure 7).

On 27 November 2023, for the formation of normotrophic scars, a cosmetologist performed a course of scar treatment with a neodymium laser from DEKA with two nozzles (7 mm 20–50 J/cm^2^ and 10 mm 40 J/cm^2^) (Figure 8).

A pharmacy preparation consisting of a heterogeneous composition of microparticles of “cross-linked” collagen of animal origin placed in a gel identical to the natural extracellular matrix was injected into the scar area (Figure 9).

At the beginning of December 2023, the patient was readmitted to the maxillofacial and plastic surgery department of the hospital for a planned procedure to treat the following diagnosis: chronic post-traumatic osteomyelitis of the maxilla. Osteomyelitis of the maxilla was determined clinically; the alveolar processes of the upper jaw were exposed, the mucous membrane was hyperemic and a gaping yellow-gray bone was detected. Foci of destruction of the upper jaw were identified using CBCT.

Osteonecrectomy was performed on 13 December 2023. The patient was prescribed 2.0 mL of the antibiotic Ceftriaxone intravenously for 7 days. The previously installed micro-screws and micro-plates fixed to the maxillary tuber on the right side were removed. Ultrasonic surgical apparatus was used to remove the nonviable parts of the maxilla. Ribbon gauzes impregnated with iodoform were placed on the formed bone wounds. The latter were sutured to the oral mucosa (Figure 10).

To date, the patient has completely lost the ability to chew and has a marked facial expression disorder on the right side of their face. The next stages of treatment will be masticatory function repair and the rehabilitation of the affected facial nerve. A peroneal graft will be transplanted to the midface to replace the missing alveolar process of the maxilla to support further prosthetics with dental implants (Figure 11).

Based on the condition of the upper limb, the consolidating fracture of the left humerus was determined. The condition after metal osteosynthesis and second-degree post-traumatic contracture of the left elbow joint is shown. Open reduction, osteosynthesis of the left elbow joint area with Kirschner wires, and the resection of the head of the left radial bone were performed.

## 4. Discussion

According to the National Park Service, the risk of a bear attack represents 1 in 2.7 million visits to protected areas [3,11]. In this case, an unexpected encounter between a human and a bear occurred during the daytime in September when the victim was harvesting pine nuts. This resulted in the predatory animal attaching the victim. According to Bombieri, G. et al. (2019), 50% of bear attacks on humans are carried out during leisure activities in forests, such as picking berries, mushrooms, etc. Interestingly, bears attack adults in 99% of cases, of which 88% are males. In 63% of cases, the victims are unaccompanied in the forest [6]. In our case, the male victim was also alone in the forest.

In southern countries, bear attacks are most often recorded when a human is grazing than during other activities. Thus, Parchizadeh J. and Belant J.L. (2021), when analyzing data on bear and leopard attacks on humans in Iran from 2012 to 2020, summarized that more than 50% of bear attacks on humans occurred while grazing livestock and almost 1.7 times less frequently (29% of cases) during other human recreational activities. In 90% of cases, bear attacks result in various injuries to humans, and in 10% of cases, they are fatal. The authors concluded that 79% of bear attacks on humans are defensive in response to chance encounters [14].

According to different authors, the frequency of bear encounters and attacks on humans is seasonal [3,5,7,14,15,16]. According to the data obtained by Parchizadeh J. and Belant J.L. (2021), 73% of bear attacks on humans occur during spring–summer [14]. The highest number of cases is recorded from May to October. According to a research paper presented by Cimpoca, A. and Voiculescu, M. (2022), most bear sightings occur in summer (39.6% of all cases), and fewer occur in autumn (28.3% of all cases), followed by spring and winter (16.9% and 15.1% of all cases, respectively) [15]. According to Korolev A.N. (2022), 55.5% of human encounters with bears occur in August–October, while May–June accounts for 30.2% of cases [5]. According to Dai Y. et al. (2019), 48.85% of instances of bears intruding into human dwellings happen in summer, 42.38% occur in autumn, and only 5.72% happen in spring [7]. This correlates with Bombieri, G. et al. (2019)’s data on the frequency of bear attacks on humans worldwide: 48% of the incidents occur during summer [6]. Most bear attacks on humans, according to the community data (https://blog.batchgeo.com/bear-attack-statistics/ accessed date 18 February 2024), happen in August (32 attacks), and equal numbers occur in July, September, and October (28 attacks in each month), followed by June (22 attacks) and May (17 attacks) [16]. Many attacks (73–97%) occur during the daytime [6,14,17].

Because of a bear attack, a person will receive multiple injuries, necessarily including the head [10,11,17,18,19,20,21]. The cranio-maxillofacial region is the most common and vulnerable site affected during a bear attack [12]. Thus, we found confirmation of the bloodthirstiness of bears in Robert Armitage Sterndale’s book “Mammalia of India” (1884) in the following description: “the victim being often terribly disfigured even if not killed, as the bear strikes at the head and face” [22].

In this clinical case, the victim was found to have multiple bite injuries to the face; fractures of the nasal bones, zygomatic bone, and maxilla; detachment of the upper rectus muscle of the left eye; a ruptured Stenon’s duct; paresis of the facial nerve; and a scalp wound in the occipital region. According to the data presented by Rasool A. et al. (2010), among the body injuries of bear attack victims, 80.57% are facial injuries, 54.67% are head injuries, and all victims have bite and laceration wounds in various localizations [17]. According to Kar I.B. et al. (2016), facial injuries occurred in 80 (57%) victims. A total of 45% of fractures of the facial bones include injuries to the mandible, 35% include injuries to the zygomatic bone, 20% include injuries to the maxilla, and 10% include injuries to the nasal bones [19]. According to Vishal et al. (2022), the zygomatic orbital complex (66.6%), mandible (40%), and maxilla (27%) are most affected [12]. Research conducted in Nepal in 2015 showed that most of the victims had injuries to the facial bones, among which the zygomatic bone was injured more often than the others [23]. A total of 15% of victims have damage to the eyeballs, which subsequently leads to visual impairment or loss [19]. Meanwhile, according to Ghezta N.K. et al. (2019), fractures of midface bones account for 71% and mandible bones account for 24% of all facial skull fractures [20]. It is possible that bears injure the faces of their victims because of the odor emanating from exposed body parts and the greater temperature of oral tissues than in the other, less-vascularized external body parts. This feature may reflect the behavior of a hungry predator. The literature indicates that bear attacks are topically different from attacks by other wild animals in that the face is the most common target [12]. This is consistent with the findings of Maurer M. et al. (2023), who noted the prevalence of trauma to parts of the face, such as the perioral and periorbital regions, ears, and nose [21].

As a result of bear attacks, the patients require long-term aesthetic rehabilitation, according to Vishal et al. (2022), disfiguring scars are found in 93% of patients. Postoperative facial deformity occurs in 80% of patients. The eyelids and nose may be affected in 20% of patients, and the cheeks are damaged in 40% of patients [12]. According to Rayamajhi S. et al. (2015), 64.71% of patients need to undergo reoperation [23].

## 5. Conclusions

Multiple injuries to the facial part of the skull lead to the formation of large wound gates that are prone to infection and complicate the visual assessment of the scale of damage and access to interventions under anesthesia.

A comprehensive approach to treatment tactics in cases of extensive injuries of the bone and soft tissue structures of the maxillofacial region significantly reduces the patient’s rehabilitation time and further increases the chances of the complete elimination of both aesthetic and functional disorders.

## Figures and Tables

**Figure 1 healthcare-12-00542-f001:**
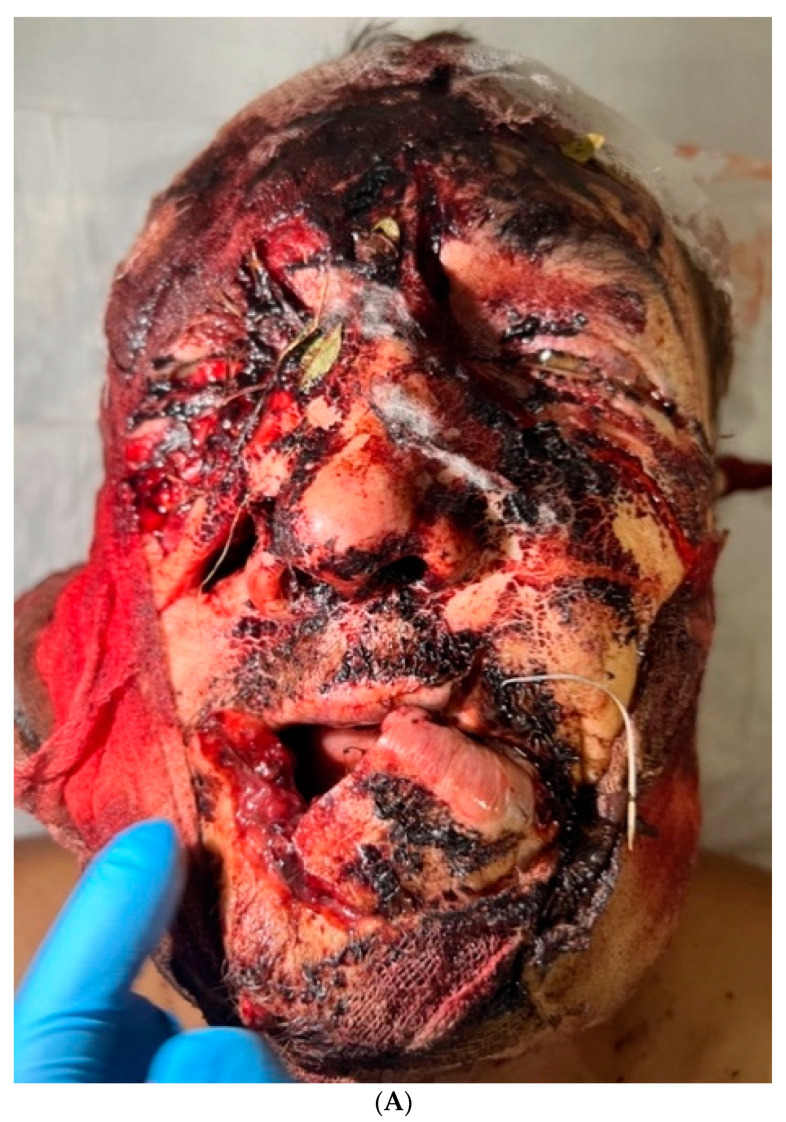
(**A**) Wounds of soft tissues of the face; (**B**) spiral computed tomography of the skull bones at the time of admission; and multiple splinter fractures of the bones of the displaced facial skeleton.

**Figure 2 healthcare-12-00542-f002:**
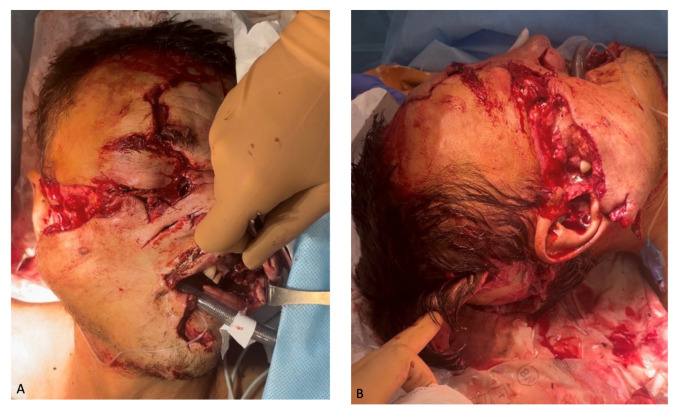
Face after initial debridement: (**A**) anterior view; (**B**) lateral view; (**C**) occipital surface view; (**D**) foreign bodies removed in the process of the wound cleaning.

**Figure 3 healthcare-12-00542-f003:**
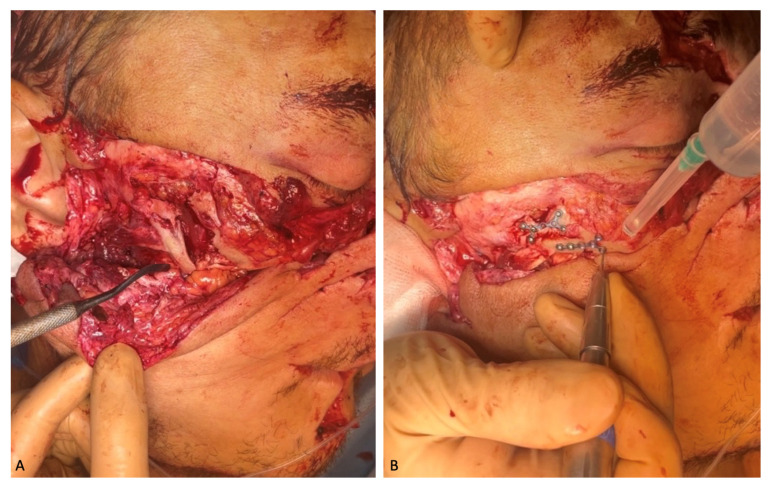
Right zygomatic bone: (**A**) preoperative osteosynthesis; (**B**) fixation of the fragments with miniplates.

**Figure 4 healthcare-12-00542-f004:**
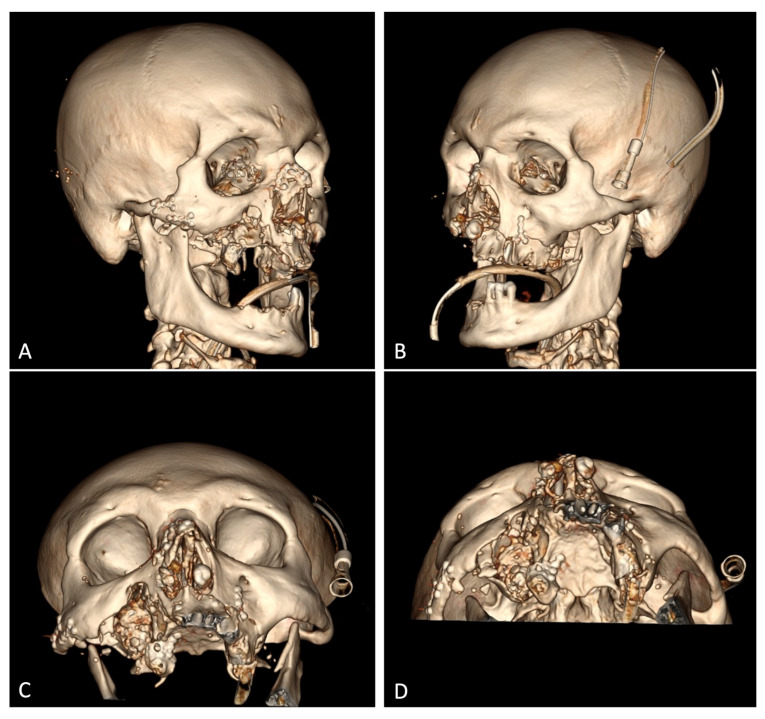
Skull spiral CT after repositioning and osteosynthesis: (**A**) right oblique-coronal projection; (**B**) left oblique-coronal projection; (**C**) anterior semi-axial projection; (**D**) axial projection.

**Figure 5 healthcare-12-00542-f005:**
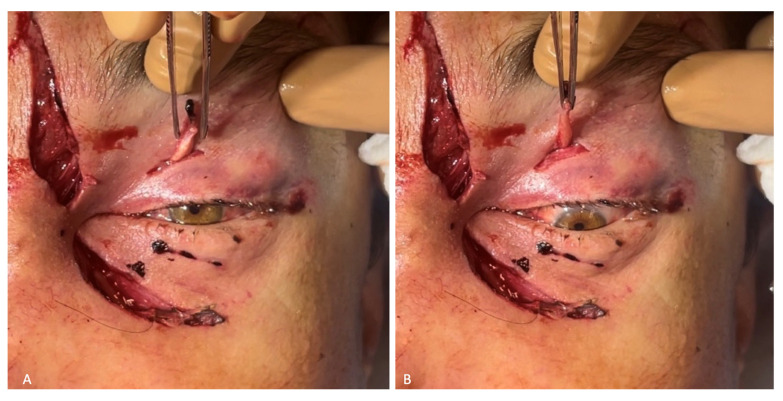
Wounds of the left periorbital region: (**A**) stump of the superior rectus muscle at rest, with pupil looking forward; (**B**) stump of the superior rectus muscle tensed, with pupil looking downward.

**Figure 6 healthcare-12-00542-f006:**
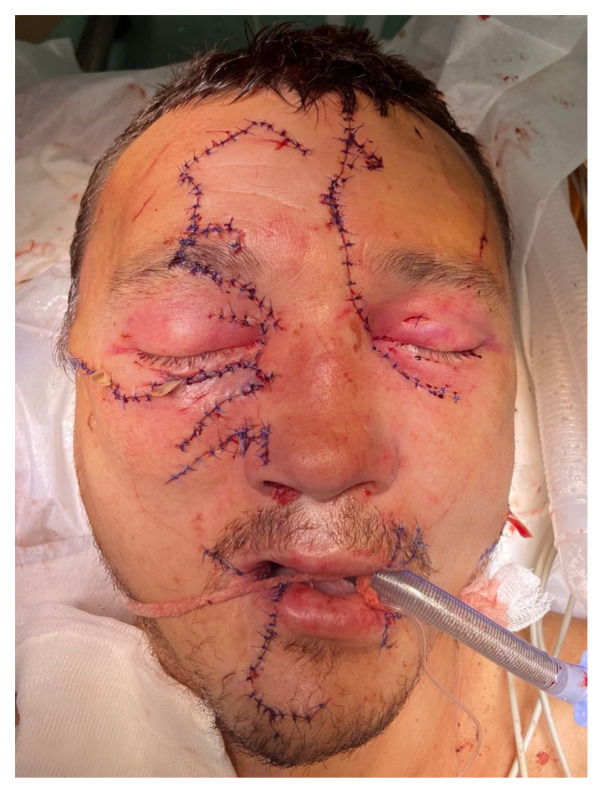
General appearance of the patient after the initial surgical debridement, repositioning, fixation of the facial bones, and suturing.

**Figure 7 healthcare-12-00542-f007:**
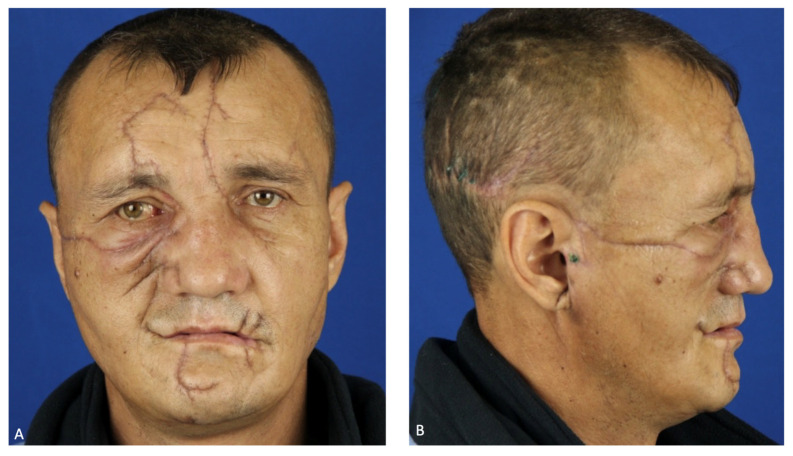
Patient’s face after 1 month: (**A**) full-face view; (**B**) profile view.

**Figure 8 healthcare-12-00542-f008:**
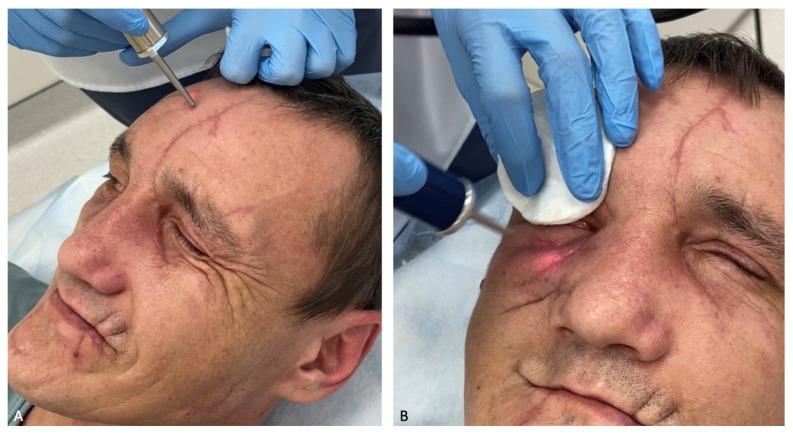
(**A**) Treatment of a scar in the frontal region with a neodymium laser with a 7 mm nozzle 20–50 J/cm^2^; (**B**) neodymium laser treatment of a scar in the suborbital region with a 10 mm 40 J/cm^2^ neodymium laser nozzle.

**Figure 9 healthcare-12-00542-f009:**
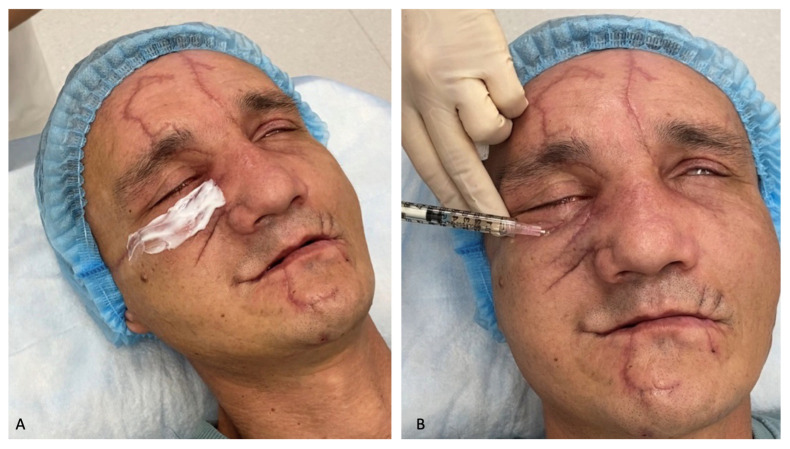
(**A**) Application of anesthesia with Acriol Pro cream (Akrihin, Staraya Kupavna, Russia); (**B**) administration of SPHEROgel Medium (YUMA Pharm, Russia).

**Figure 10 healthcare-12-00542-f010:**
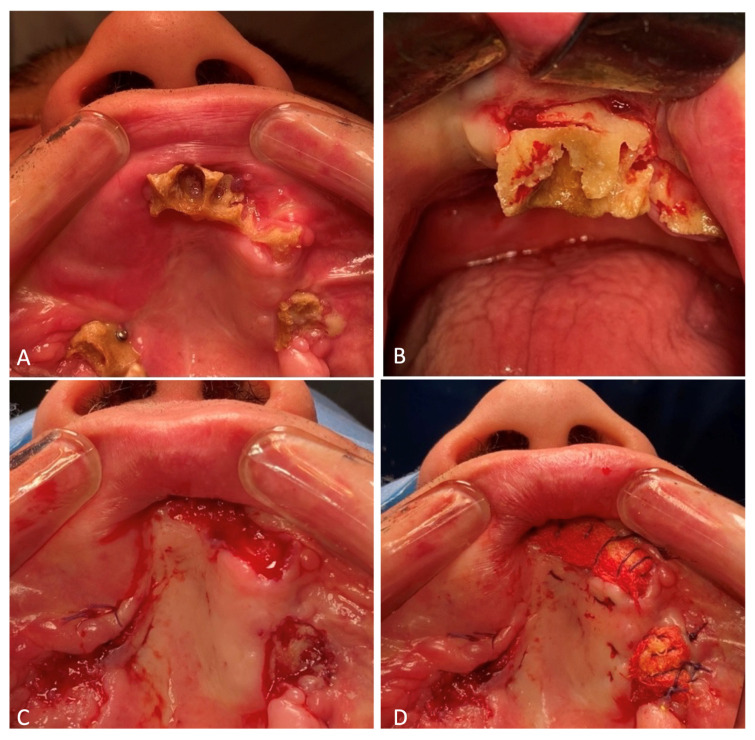
Osteonecrectomy of the maxilla: (**A**) necrotized alveolar processes of the maxilla; (**B**) frontal section of the alveolar process of the maxilla during sawing; (**C**) view of the maxilla after necrectomy was performed; (**D**) palatine surface of the maxilla with iodoform ribbon gauze sutured.

**Figure 11 healthcare-12-00542-f011:**
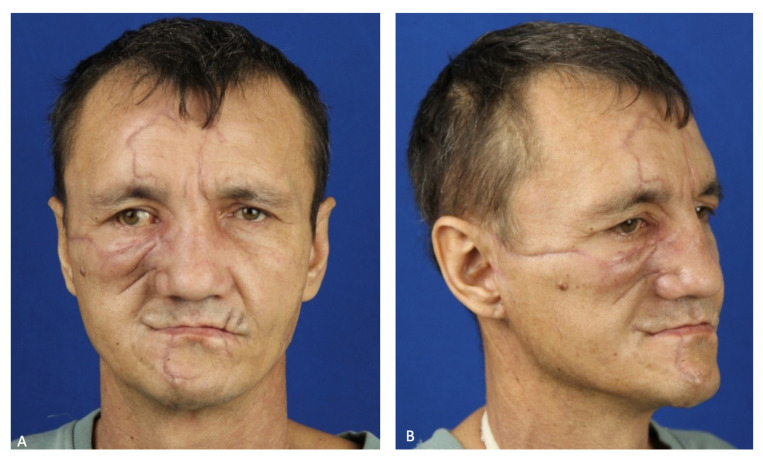
Patient’s face after 2.5 months: (**A**) full-face; (**B**) 2/3 view.

## Data Availability

Data are contained within the article.

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
