# Peer review of "Polytrauma Caused by a Bear Attacking a Human with a Benign Outcome"

_healthcare, 2024, doi:10.3390/healthcare12050542_

Round 1

Reviewer 1 Report

Comments and Suggestions for Authors TLine 4 : Line 4 : Line 4 : Line 4 :

Line 4 : which year in september-december) ? 

Line 33-38 : Line 33-38 : Line 33-38 : Line 33-38 :

Line 33-38 : please add reference

change materials and change materials and change materials and change materials and

change materials and methods to Case presentation 

Line 52 : Name of institution should be added

line 56: exact day should not mentioned as it can leads to patients identity 

Line 68 : which type of fracture in upper limb ? how was treated?

any empiric antibiotic was administred in emergency? 

Remove results subheading 

all steps of surgical treatment was done at what duration , was there any infection during stages ? any antibiotic ? if yes what type and for how long

was there any bone gap in which a vasculrized fibula was needed ?

Line 239 : osteomyelitis of maxilla should be detailled , how was it diagnosed, what imaging test/ clinical signs /biological test, was union achieved? 

was informed written consent was taken from patient ? this should be included in the main text

challanges of case challanges of case challanges of case challanges of case

challanges of case should be detailed in the discussion specially osteomyelitis  (type of bacteria, medical and surgical treatment for OM) 

overall this is very interesting study and merit for publication, i congratulate authors for this work and patient outcome 

Author Response

Dear reviewer!   We express our gratitude for your attention to our work. Thanks to you, we were able to significantly improve the quality of data presentation and hope that you will be satisfied with the result   Below we provide links to the lines in which the changes were made   Line 10: added year September – December 2023 Line 89-90: The patient signed an informed written consent, which included permission to publish the medical history data and photo protocol Line 32: added reference, updated reference numbering: [2] 2. Støen OG, Ordiz A, Sahlén V, Arnemo JM, Sæbø S, Mattsing G, Kristofferson M, Brunberg S, Kindberg J, Swenson JE. Brown bear (Ursus arctos) attacks resulting in human casualties in Scandinavia 1977-2016; management implications and recommendations. PLoS One. 2018 May 23;13(5):e0196876. doi: 10.1371/journal.pone.0196876. Line 65: added full hospital name line 69: specific date and time removed, replaced with “morning of September 2023” Lines 153-155: added description of upper limb injury Lines 154-255: Remove results subheading Lines: we have added a description of the clinical picture of osteomyelitis and treatment Lines 157-159: describes the treatment of upper extremity trauma during initial hospitalization and on lines 277-281 during readmission. Lines 164-165: Antibacterial therapy described Lines 227-228: antiseptic therapy for salivary fistulas described   We would like to answer the question about restoring the violation of the integrity of the upper jaw with a fibular graft. In an emergency situation, it is not carried out due to several factors: the presence of injury, the severity of the patient's condition (hemorrhagic and painful shock), the presence of inflammation, foreign infection, it is impossible to reliably determine the volume of reconstruction. In the future, one of two ways to restore chewing function will be chosen: 1 reconstruction with a vascularized fibular, iliac crest or rib graft. 2 – installation of zygomatic implants followed by prosthetics on them.

Reviewer 2 Report

Comments and Suggestions for Authors

The article "Polytrauma caused by bear attack on a human with a benign outcome" is a detailed case report documenting the medical management of a patient who suffered multiple traumas from a bear attack. It highlights the rarity of such incidents in the literature despite their high frequency in Siberia due to the presence of brown bears. The case presents a comprehensive treatment journey from emergency care to rehabilitation, including maxillofacial and reconstructive surgery, and the use of innovative treatments like neodymium laser therapy and injections of "cross-linked" collagen for scar management.

- **Strengths**: The article provides a thorough and insightful analysis of a rare medical case, contributing valuable information to the medical community about handling complex wildlife attack injuries. The detailed description of the surgical procedures and postoperative care, including innovative treatments, is informative and demonstrates a multi-disciplinary approach to trauma care.

- **Weaknesses**: The article could benefit from a broader discussion on preventive measures and public health implications to reduce bear-human encounters, which are significant in areas where such attacks are more common. Additionally, while the case is well-documented, the article might have enhanced its impact by including a review of similar cases to provide a comparative analysis of treatment outcomes.

- **Suggestions for Improvement**: Future editions could expand on the statistical data regarding bear attacks globally and regionally, offering readers a clearer understanding of the prevalence and risk factors associated with such incidents. Incorporating patient follow-up data would also enrich the article, providing insights into the long-term efficacy of the treatment modalities used.

Overall, this article is a valuable addition to the existing body of knowledge on managing severe trauma cases resulting from wildlife attacks, showcasing the complexity and interdisciplinary approach required for successful patient outcomes.

Author Response

Dear reviewer!   We thank you for your review and the work you have done in evaluating our research. Thanks to your valuable comments and recommendations, we have been able to greatly expand the introduction and discussion.   Below we provide links to the lines in which the changes were made In the introduction, lines 51-60: added information about the treatment of patients with multiple injuries after a bear attack, as well as features of the microflora of animals. In the discussion on lines 293-300, information about the nature and characteristics of bear attacks on humans has been added. On lines 302-303 it is written about the seasonality of attacks. Lines 319-323: We added confirmed historical facts of bear attacks, and also indicated the nature of the damage. Lines 332-335, 342-343, 346-350 indicate the most common lesions of the facial areas.